# Quantitative PCR (qPCR) assay for the specific detection of the Chinese mystery snail (*Cipangopaludina chinensis*) in the UK

Helen C. Rees[1]*, Gavin H. Measures[2], Steven D. Kane[1], Ben C. Maddison[1]

**1** Biotechnology, RSK ADAS Ltd, Nottingham, United Kingdom, **2** Species Recovery and Reintroductions Team, Natural England, Peterborough, United Kingdom

\* helen.rees@adas.co.uk

**Data Availability Statement:** All the relevant data are within the paper and its Supporting Information files.

## Abstract

*Cipangopaludina chinensis* Gray 1833 is an East Asian freshwater snail and invasive species in many parts of the world (Global Invasive Species Database, 2022). Within the UK, it was first found at the Pevensey Levels, Sussex, and has since been reported at a second site at Southampton Common, Hampshire. Both sites are designated as Sites of Special Scientific Interest (SSSI) for their wildlife importance. Although the impacts of this species within the UK have not yet been investigated several exotic parasites of the snail have been reported and research suggests that its presence can negatively impact native snail species. This is especially important at the Pevensey Levels due to the presence of the rare freshwater mollusc *Anisus vorticulus* (Little Whirlpool Rams's-horn snail). Here, we have developed a qPCR-based eDNA assay for the detection of *C. chinensis* and compared water samples tested for eDNA with results from manual survey of the ditches at the Pevensey Levels. Our eDNA analysis exhibited an overall observed percentage agreement of 80% with a kappa coefficient of agreement between manual and eDNA surveys of 0.59 (95% CI 0.31 to 0.88). Some samples which were qPCR negative for *C. chinensis* were positive by manual survey, and *vice versa* revealing the potential for improved overall detection rates when using a combination of manual and eDNA methodologies. eDNA analysis can therefore augment manual survey techniques for *C. chinensis* as a relatively quick and inexpensive tool for collecting presence and distribution data that could be used to inform further manual surveys and control measures within the ditches.

## Introduction

Knowledge of species distribution is critical to ecological management and conservation biology. Effective management requires the detection of populations which can sometimes be at low densities and is usually based on visual detection and counting. There has been increasing interest in using environmental DNA (eDNA) techniques to monitor freshwater invertebrate communities and in particular invasive non-native species.

Environmental samples (water, soil, sediment, air etc.) contain DNA that originates from sloughed cellular material (e.g. skin cells); DNA that is excreted (e.g. faeces and urine); or

**Funding:** This project was funded by a grant to HR from Natural England. Award reference: Chinese mystery snail, Pevensey Levels Sussex. GM of Natural England played a role in the design, sample collection, and preparation of manuscript. https://www.gov.uk/government/organisations/natural-england.

**Competing interests:** NO authors have competing interests

**Abbreviations:** COI, Cytochrome oxidase I; eDNA, Environmental DNA; qPCR, quantitative polymerase chain reaction; NNR, National Nature Reserve; SSSI, Site of Special Scientific Interest.

DNA that is secreted (e.g. saliva) by organisms occupying (or visiting) the environment in question and as such is known as eDNA [1–3]. The presence of an organism's DNA within an environmental sample such as wateris short lived and eDNA has been shown to be degraded to undetectable levels within days to weeks. [4–6]. The detection of an organism's DNA is therefore a demonstration of its presence/recent presence and can be used instead of manual survey or capture of the organism itself [7, 8].

For just over a decade eDNA has been used as a non-invasive sampling technique. It has been shown to be reliable, cost effective, less harmful to the ecosystem and correlates well with conventional survey results [9, 10]. As such, the analysis of eDNA using PCR-based approaches has become an important tool for measuring presence/absence of various aquatic species including the highly invasive mollusc *Potamopyrgus antipodarum* [11] New Zealand mud snail within rivers [12, 13].

The Chinese mystery snail *Cipangopaludina chinensis* [11] or 'trapdoor snail' is an extremely large (40-60mm high, 30-40mm wide), globose operculate snail native to East Asia and is the largest freshwater snail in Britain (Fig 1). The 'trapdoor' refers to an oval plate (operculum) which seals the aperture of the snail when the snail is fully retracted. *C. chinensis* is a problem invasive species in many parts of the world and has only relatively recently been introduced to Europe. It has been present in the Netherlands since at least 2007 [14] and was found at the Pevensey Levels, Sussex, UK in 2018 (Willing 2018 [unpublished]) and has since been reported at Southampton Common, Hampshire, UK in 2022 (Measures [unpublished]). The species is sold for ornamental ponds and aquaria and is also eaten in certain Asian communities. It is thought that either of these may have resulted in the accidental (or intentional) introduction of *C. chinensis* at the two sites in Britain resulting in breeding populations. Females can produce up to 30 live young per year [14] and live for up to five years [15] which could produce a boom in population. Across Asia, multiple species of *C. chinensis* have been recognised and there is considerable confusion on its taxonomy. For example *Cipangopaludina fluminalis* [16] is a synonym of, and *Cipangopaludina wingatei* [17] is a subspecies of *C. chinensis* [18]. Two subspecies of *C. chinensis* that are recognised are *C. chinensis chinensis* [11] and *Cipangopaludina chinensis malleata* (also known as *Cipangopaludina chinensis laeta*) [19, 20].

The Pevensey Levels site is within an internationally important area of coastal grazing marshes which is also designated as a Special Area of Conservation and a Ramsar site. At the Pevensey Levels the snail lives and breeds in a heavily vegetated shallow drainage ditch system mostly in cattle-poached margins and were first found in September 2018. Survey work carried

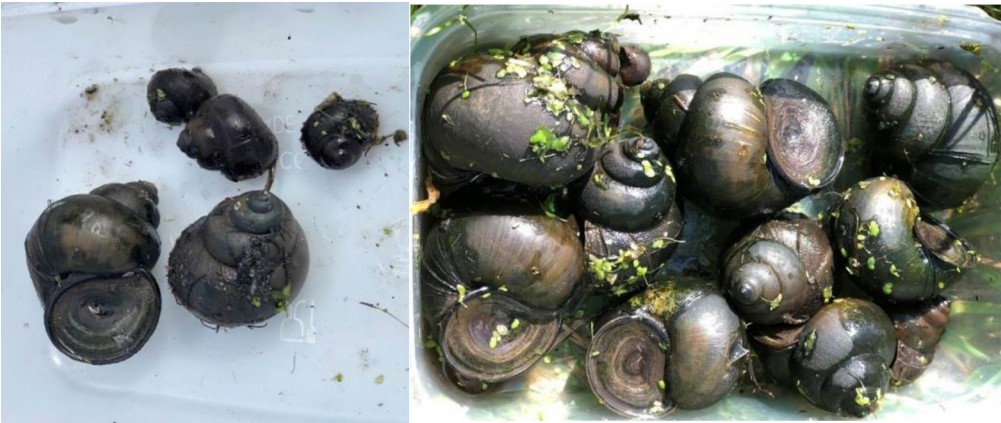

**Fig 1. Images of *C. chinensis.*** *Specimens collected from Pevensey Levels August 2021 (© Gavin Measures, NE).*

out in July 2019 found the presence of a recruiting population of the snail in an approximately 400 m stretch of the ditch and further surveys carried out at this ditch system in February 2021 found a number of smaller juveniles which suggested that a breeding population was present (Willing 2021a [unpublished]). Little is known about the impact of *C. chinensis* in Britain, however at the Pevensey Levels, there is a concern that should *C. chinensis* spread from its current locations within the ditch system then there will be a risk of disruption to the diverse and extensive freshwater ecosystems at the site. *C. chinensis* are considered to be an aquatic invasive species due to their ability to avoid predation and out-compete indigenous mollusc species for resources. Research has shown that their presence can negatively impact native snail species in mesocosm experiments [21] and several exotic parasites of *C. chinensis* have been reported [22]. Additionally, they excrete a large amount of faecal matter which can affect nitrogen and phosphorous cycling in aquatic ecosystems [23]. The ditch system in question is a habitat for many rare invertebrate species including rare the freshwater Mollusca the little whirlpool ram's-horn snail, *Anisus vorticulus* [24] and the shining ram's-horn snail, *Segmentina nitida* [25].

This study therefore set out to develop an eDNA-based assay for the detection of *C. chinensis* which could be used to inform and complement traditional manual surveys at the Pevensey Levels and the success of any eradication or mitigation operations over subsequent years. Here, we describe a qPCR assay for the detection of *C. chinensis* and demonstrate the detection of this species at this UK site.

## Materials and methods

### Sampling design

The parts of the ditch system at the Pevensey Levels that were targeted for eDNA sampling were those where there had been previous manual records for *C. chinensis*, plus connected and nearby ditches, up to a total of 36 water samples. eDNA sampling effort aimed to collect enough samples to obtain adequate representation of occupied areas of the site.

### Sample collection: Pevensey Levels 2021

36 ditch water samples were collected by Natural England and RSK ADAS staff at Pevensey Levels between the 10th and 12th August 2021 (Fig 2). All sample information can be found in S1 Table in S1 File. In addition, 32 ditch samples were collected outside the known range of *C. chinensis* (Stodmarsh, Kent and Leicestershire) and were used as assay specificity controls. These were collected by Natural England and RSK ADAS staff between the 16th January and the 27th November 2020 and 10th January and the 4th February 2021 respectively. All sample information can be found in S2 and S3 Tables in S1 File). Each sample was collected using its own dedicated sampling pack i.e. sterile disposable gloves, a sterile scoop, a sterile sampling bag, and a sterile 50 mL syringe. 20x 30 mL water samples were collected along the length of each sampling site at the ditch system and pooled into a sterile sampling bag. Up to 500 mL of this water was filtered through a 0.22 μm PES sterivex filter (Millipore; product number SVGP01050) using a 50 mL luer-lock syringe (Table 1). After filtration of water, 95% ethanol was added as a preservative. One control field blank (distilled water) was also filtered at the Pevensey Levels ditch system.

### Sample collection: Pevensey Levels 2022

18 ditch water samples were collected at Pevensey Levels on the 3rd August 2022 (Fig 3). All sample information can be found in S4 Table in S1 File. Two control field blanks (same volume of distilled water) were also filtered by ADAS staff next to sampling site 8.

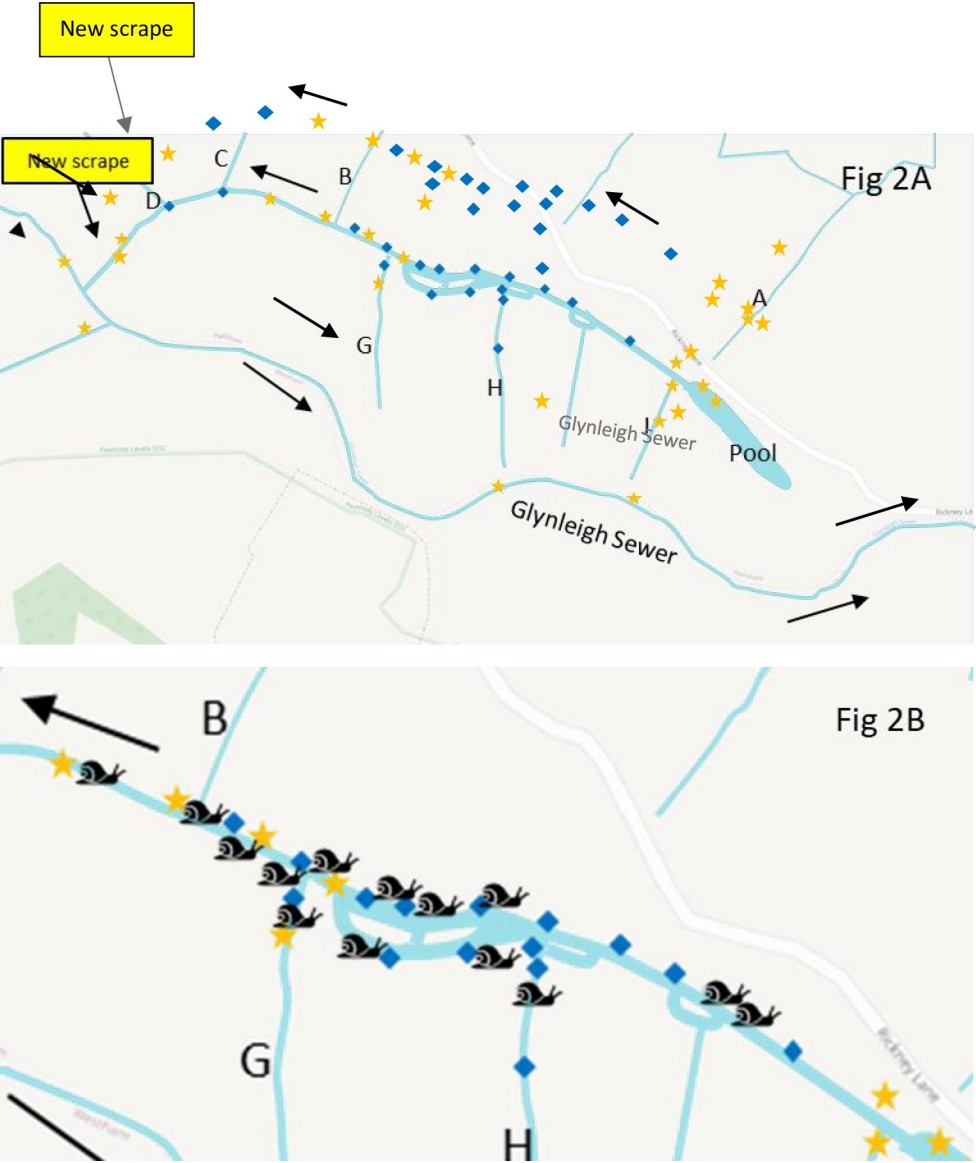

**Fig 2.** A) Ditch system at Pevensey Levels showing *C. chinensis* DNA results, August 2021. B) Ditch system at Pevensey Levels showing *C. chinensis* manual and DNA survey results, 2020–2021. Blue diamonds and orange stars mark the eDNA sampling sites and whether positive or negative for *C. chinensis* DNA, respectively. Snail icons mark the presence of C. chinensis specimens found during manual survey. Black arrows show direction of flow of main ditch. Side ditches are labelled A-J. OpenStreetMap© (data available under the Open Database License).

For all samples two litres of water was collected along the length of each sampling site at the ditch system and pooled into a sterile sampling bag to allow for two types of filters to be tested for their ease of use and filterable volume of water. As in the 2021 eDNA survey a 0.22 μm Ster-ivex polyethersulfone (PES) filter (Merck Millipore, Darmstadt, Germany) was used alongside a second filter type: 0.45 μm PES filter (GVS, Stanford, North America) in combination with a 3.1 μm glass fibre pre-filter (GVS, Stanford, North America) (recommended where water to be sampled is turbid—used to remove larger particles of sediment and plant material before cap-turing eDNA on the main filter). Up to 500 mLs of water was filtered and 95% ethanol was added as a preservative.

**Table 1. Summary of *C. chinensis* survey and PCR status of the 36 water samples collected from the Pevensey Levels in 2021.**

| Sampling Site | Volume filtered (ml) | DNA (ng/µL) | qPCR results (2021) | Manual Survey result (2021) |
|---|---|---|---|---|
| **EASTERN SECTION DITCH** | | | | |
| 1–3 pool—1 | 230 | 3.71 | 0/12 | Not found |
| 1–3 pool—2 | 150; 150* | 0.88 | 0/12 | Not found |
| 4 | 130 | 2.74 | 0/12 | Not found |
| 5 | 250 | 2.78 | 4/12 | Not found |
| 6 | 260 | 4.68 | 11/12 | Not found |
| 6–7 | 400 | 4.12 | 11/12 | Present |
| 7 | 100; 100* | 5.74 | 12/12 | Present |
| 8 | 500 | 5.29 | 0/12 | Present |
| 8–18 lower side | 500 | 13.50 | 12/12 | Present |
| 8–18 upper side | 200 | 21.40 | 12/12 | Present |
| 18 | 300 | 2.88 | 12/12 | Present |
| 18–7 lower | 400 | 13.30 | 12/12 | Present |
| 18–7 upper | 250 | 23.30 | 1/12, 3/12, 4/12[#] | Present |
| **WESTERN SECTION DITCH** | | | | |
| 12 | 120; 100* | 4.61 | 11/12 | Present |
| 17–16 | 170; 170* | 21.80 | 0/12 | Present |
| 16 | 150; 150* | 11.20 | 0/12 | Present |
| End of barn at oak tree (near 16) | 150 | 4.66 | 1/12, 0/12, 2/12[#] | Present |
| 15 | 90; 80* | 8.88 | 0/12 | Present |
| 11 opposite ditch C | 250 | 4.48 | 1/12, 3/12, 2/12[#] | Not found |
| 9 opposite ditch D | 300 | 3.01 | 1/12, 4/12, 4/12[#] | Not found |
| 13 opposite scrape in field | 140; 140* | 10.90 | 0/12 | Not found |
| **SIDE DITCHES** | | | | |
| 4-J (between 4 and ditch J) | 350 | 2.77 | 0/12 | Not found |
| Ditch J | 265 | 1.49 | 0/12 | Not found |
| Ditch A | 230 | 4.95 | 0/12 | Not found |
| Ditch G lower (near 12) | 70; 80* | 5.41 | 2/12, 4/12, 5/12[#] | Present |
| Ditch G upper above culvert | 290 | 7.08 | 0/12 | Not found |
| Ditch H lower section (near 7) | 250 | 2.18 | 4/12 | Present |
| Ditch H higher section | 280 | 15.60 | 2/12, 2/12, 3/12[#] | Not found |
| **WIDER DITCH NETWORK** | | | | |
| Down Sewer Main | 300 | 1.37 | 0/12 | N/S |
| Down Sewer—small ditch opposite | 250 | 5.41 | 0/12 | N/S |
| Horse Eye Sewer | 250 | 2.11 | 0/12 | N/S |
| Rickney Sewer | 120; 150* | 2.54 | 0/12 | N/S |
| Site L (Marland Sewer) | 260 | 2.89 | 0/12 | N/S |
| **GLYNLEIGH LEVEL SEWER** | | | | |
| Main sewer F upstream of main ditch | 400 | 1.02 | 0/12 | Not found |
| Main sewer K downstream of ditch J | 300 | 2.01 | 0/12 | Not found |
| Main sewer I | 300 | 3.58 | 0/12 | Not found |

Green highlighted samples were found to be positive for *C. chinensis* by both qPCR and manual survey; orange highlighted surveys were found to be positive for *C. chinensis* by qPCR but not manual survey; blue highlighted samples were found to be positive for *C. chinensis* by manual survey but not by qPCR.

N/S denotes those sites where manual survey did not take place.

* Samples were more than one sterivex filter had to be used due to filter clogging.

[#] qPCR was repeated for these samples to confirm positivity.

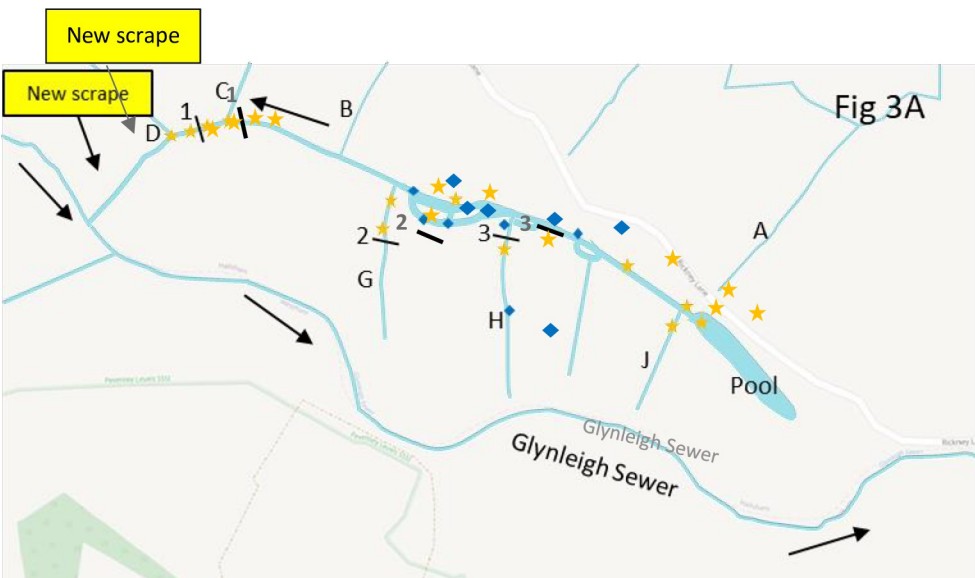

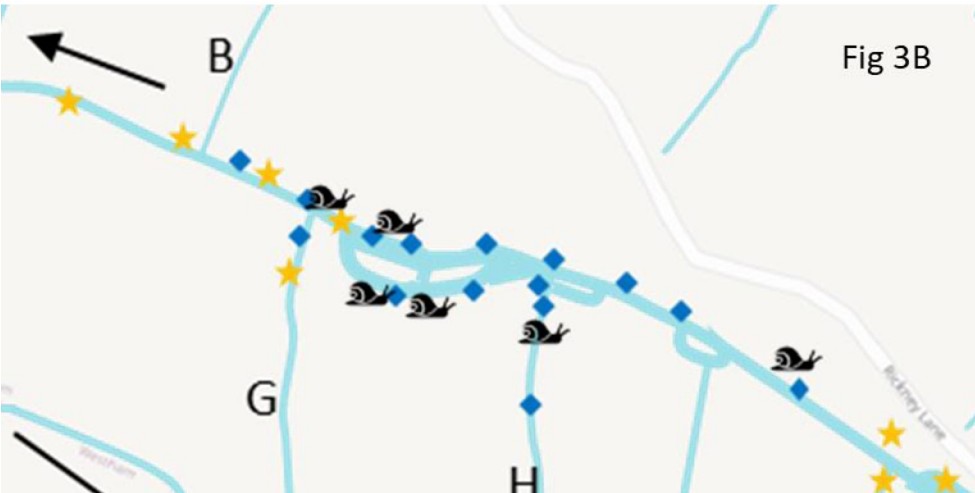

**Fig 3.** A) Ditch system at Pevensey Levels showing *C. chinensis* DNA results, August 2022. B) Ditch system at Pevensey Levels showing *C. chinensis* manual and DNA survey results, 2020–2021. Blue diamonds and orange stars mark the eDNA sampling sites and whether positive or negative for C. chinensis DNA, (respectively). Snail icons mark the presence of C. chinensis specimens found during manual survey. Many of the sites in the main infested ditch were not eDNA or manual surveyed in 2022 as were assumed to be positive for *C. chinensis*. Side ditches are labelled A-J. Black lines indicate the sites of the three coffer dams installed in 2021/2022 (numbered 1 to 3 from left to right). OpenStreetMap© (data available under the Open Database License).

## DNA extraction

Upon return to the laboratory, the preservative solution was removed from the filters and any DNA containing material captured on the filter membrane was recovered by addition of lysis buffer and proteinase K from the DNeasy blood and tissue kit (Qiagen) to the filter as follows:

1. For sterivex filters– 760 μL of ATL buffer/proteinase K (20:1 ratio) was added to each filter.

2. For the PES/GF filter combination– 580 μL ATL buffer/proteinase K (20:1) was added to each filter.

The ends of the filters were sealed and briefly agitated by vortexing before incubation overnight at 37˚C. The supernatant was then extracted using the DNeasy blood and tissue kit (Qiagen) following the manufacturer's instructions with final resuspension in 200 μl of elution buffer. All DNA samples were quantified using the Qubit® dsDNA BR assay kit and Qubit 3.0 Fluorometer (Invitrogen) following the manufacturer's instructions then stored at -20 ℃ before use. Field blanks (x1) and extraction blanks (x2) were also processed alongside the Pevensey Levels sterivex filters. All eDNA extractions from filters were performed in a separate laboratory remote from snail specimen DNA extraction and qPCR set up, using dedicated tissue extraction equipment. Disposable laboratory coats were worn, and benches and equipment were wiped down with a 10% bleach solution before and after use.

## Specimen collection and DNA extraction

Snail specimens of both *C. chinensis* and native species were collected from ditches at the Pevensey Levels site by hand netting using a survey net fitted with a 2 mm mesh by Natural England staff in August 2021 [26]. Specimens were preserved in 95% ethanol prior to couriering to the RSK ADAS laboratories.

Each snail specimen was individually transferred to a clean, sterile petri dish and a photographic record was taken. For larger specimens a fresh sterile scalpel blade was used to remove a piece of tissue from each specimen and macerate the tissue into a fine paste before placing into a sterile 1.5 mL Eppendorf tube. After use mortar and pestles were immediately immersed in 10% bleach for a minimum of 10 minutes and then cleaned in between samples with 10% Distel (Tristel™), rinsed with dH₂O and then autoclaved at 121 ℃ for 15–20 minutes. For smaller specimens the whole organism was transferred into a sterile 1.5 mL Eppendorf tube and using a fresh sterile Eppendorf pestle was ground into a fine paste.

DNA was extracted from snail specimens using DNeasy blood and tissue kit (Qiagen) following the manufacturer's instructions: 200 μL ATL and 20 μL PK buffer was added to each specimen and incubated for 1–2 hours at 56˚C until completely digested. Final resuspension was in 100 or 200 μL AE buffer for specimens. Extracted DNA was quantified using the Qubit® dsDNA BR assay kit and Qubit 3.0 fluorimeter.

## Specimen identification PCR

All PCR set up was performed in a clean 'PCR room' within a UV sterilisable cabinet within a separate laboratory to DNA extraction using dedicated equipment and PPE. To ensure the unidirectional workflow DNA extracts are collected from the DNA extraction laboratory and transferred to the PCR set-up laboratory.

DNA barcoding was performed to confirm the identity of the provided specimens using the COI primers mICOIintF 5'-GGWACWGGWTGAACWGTWTAYCCYCC-3' and jgHCO2198 5'-TAIACYTCIGGRTGICCRAARAAYCA-3' [27, 28]. After PCR and amplicon clean-up using the Machery-Nagel™ NucleoSpin™ Gel and PCR Clean-up Kit according to the manufacturers instructions, PCR products were Sanger sequenced and returned sequences identified using BLAST.

## Species specific primer design and validation

To design primers specific to *C. chinensis* the DNA sequences for the cytochrome oxidase 1 (COI) gene for *C. chinensis* and other closely related and/or co-occurring snails commonly found at the Pevensey Levels and the wider UK were downloaded from BOLD [29] and Genbank [30] and their sequences aligned using BioEdit version 7.2.5. This list of species comprised the 54 species of freshwater snails known to occur in the UK, both native and non-

native [31]. Primers and probes were designed using PrimerBLAST [32] with default settings. Ten potential primer/probe combinations were generated and reduced to four using Primer-BLAST and looking for cross-species amplification.

The four potential primer/probe combinations were tested firstly on DNA extracted from *C. chinensis* followed by 16 other co-occurring snail species to test for cross-species reactivity (S5 Table in S1 File). PCRs were set up in a total volume of 25 μL consisting of: 1 μL of each primer/probe (final concentrations: 0.2 μmol/L forward primer; 0.8 μmol/L reverse primer; 0.5 μmol/L probe), 12.5 μL of TaqMan® Environmental Master Mix 2.0 (containing Ampli-Taq GOLD DNA polymerase), and 6.5 μL ddH2O. 3μL of DNA was added to each reaction and 12 replicates were performed per DNA sample. All controls were performed in quadrupli-cate with a dilution series of $1\times10^{-1}$ to $1\times10^{-4}$ ng/μL *C. chinensis* DNA being used as positive controls and ddH2O in place of DNA for the negative control. PCRs were run on a Bio-Rad CFX Connect real-time PCR machine as follows: an initial incubation for 5 minutes at 56.3 ℃ then 10 minutes at 95˚C; followed by 55 cycles with a melting temperature of 95˚C for 30 sec-onds and an annealing temperature of 56.4 ℃ for 1 minute.

Once specificity of primer/probe combinations was confirmed, the most promising primer primer/probe combination (no cross reactivity, highest RFU, best shaped curve) was optimised by independently varying final primer concentrations (the probe was held at a final concentra-tion of 0.1 μmol/L) [33]. The sensitivity of the assay was tested by creating a seven-level stan-dard curve dilution series (1 ng/μL to $1\times10^{-6}$ ng/μL). 12 replicates of each dilution were run using the optimised primer/probe concentrations to determine the limit of detection (LOD) and limit of quantification (LOQ).

The optimised assay using primer/probe combination 8 (CMS 8F: 5'- GTGATTGTAACT GCTCACGCAT-3'; CMS 8R: 5'- CCCAACACCCCCTTCAACAG-3'; CMS probe: 5'- GG AGCTCCTGATATAGCTTTTCCTCGT-3', FAM/BHQ-1) was used to determine the presence/absence of *C. chinensis* within the 36 ditch samples from the Pevensey Levels and the 30 ditch sam-ples from outside the known range of *C. chinensis* (S2 and S3 Tables in S1 File). A selection of posi-tive amplification were Sanger sequenced and returned sequences and identified using BLAST.

## Inhibition testing

All DNA extracts were tested for inhibition by adding 3 μL of extract into a qPCR for the amplification of an internal positive control (DNA derived from the plasmid pSD3 previously used in our laboratories). PCRs were set up in a total volume of 25 μL consisting of: 3 μL of internal positive control DNA (0.08 ng/μL),1 μL of each primer/probe (final concentrations: 0.2 μmol/L forward primer; 0.4 μmol/L reverse primer; 0.1 μmol/L probe; S2 File), 12.5 μL of TaqMan® Environmental Master Mix 2.0, and 3.5 μL ddH2O. All positive and negative con-trols were performed in triplicate with a dilution series of 0.08 ng/μL to 0.0032 ng/μL inhibtion control DNA being used as positive controls and ddH$_2$O in place of DNA for the negative con-trol. PCR cycling was as above.

## Statistical analysis

To measure the agreement between the two survey methods, that is, manual survey and eDNA analysis, Cohen's kappa coefficient [34] was calculated as follows:

$$k = \frac{\Pr(a) - \Pr(e)}{1 - \Pr(e)}$$

where Pr(a) is the relative agreement among rates, and Pr(e) is the hypothetical probability of chance agreement, using the observed data to calculate the probabilities of each method

randomly giving a positive detection. If the methods are in complete agreement, then j = 1. If there is no agreement other than what would be expected by chance, j = 0. Kappa analysis did not include samples taken from outside the known range of *C. chinensis* or any other samples taken from sites that had not been manually surveyed.

Once PCR analyses had been performed, the program PRESENCE version 2.13.43 [available from http://www.mbr-pwrc.usgs.gov/software/presence.html [35]] was used for occupancy modelling of the data. A single-season model was used which assumes that species are never falsely detected at a site when absent, but that may or may not be detected when present; the detection of a species at an individual site is independent of the detection of the species at all other sites; and the probability of detecting the species across all sites is constant.

## Results

### Detection

Snail specimens were identified by Sanger sequencing followed by BLAST analysis (S5 Table in S1 File) and a subset chosen for assay specificity testing. Four primer/probe combinations were found to be species-specific for *C. chinesis* in that no cross-amplification was detected when tested with DNA isolated from various snail specimens collected at the Pevensey Levels and Stodmarsh NNR. Primer/probe combination 8 was found to be the most promising, was optimised and gave a limit of detection (LOD) and limit of quantification (LOQ) of $1\times10^{-4}$ ng/μL (S2 File). Here we define LOD as the lowest standard concentration at which 95% of technical replicates amplify and LOQ is the lowest standard concentration for which the coefficient of variation (CV; equal to the standard deviation quantity divided by the mean quantity of a group of replicates) value is <35% [36]. Detection in a water sample was indicated by at least 1 of 12 positive qPCR replicates [9] however, all 1 of 12 results were confirmed by retesting twice more and samples were only considered to be positive if one or both of the retests showed any positive amplifications, those that were negative on both retests were considered to be negative for *C. chinensis* DNA. It should be noted that during any subsequent analysis samples went through an additional round of freeze-thawing which can lead to DNA degradation and therefore a loss of eDNA detection. This could in part account for the sample now being negative although it should also be noted that three other samples were still positive for *C. chinensis* DNA when retested despite also going through an additional round of freeze-thawing. None of the samples caused inhibition of the qPCR and all the field, extraction, and PCR blanks were negative for *C. chinensis* DNA.

The optimised assay for *C. chinensis* was used on the DNA extracted from all Pevensey Levels ditch water samples and ditch samples from outside the known range of *C. chinensis*. In 2021, 16 out of the 36 Pevensey Levels samples were found to be positive for *C. chinensis* DNA (Table 1, Fig 2). When comparing the two filter types during the 2022 Pevensey Levels sampling, the 0.45 PES/GF filter combination found 6 out of the 20 samples to be positive whilst the sterivex filers only found two out of 20 samples to be positive for *C. chinensis* DNA (Table 2, Fig 3). All remaining samples were negative for *C. chinensis* DNA, therefore other than any ditches where *C. chinensis* was found by manual survey, *C. chinensis* is likely to be absent from these ditches (Tables 1 and 2). It was also noted that fewer live *C. chinensis* were observed during the eDNA surveys in 2022 than were observed in the eDNA surveys in 2021, although a number of empty shells were seen in 2022 at both sites.

In order to confirm the presence of *C. chinensis* a selection of positive amplifications were Sanger sequenced and analysed using BLAST. All the positive amplifications sequenced were determined to be *C. chinensis laeta* as were most of the collected *C. chinensis* specimens with (S5 Table in S1 File).

**Table 2. Summary of *C. chinensis* survey and PCR status of the water samples collected from the Pevensey Levels in 2022.**

| Sampling site | Volume filtered (mL) | | DNA (ng/µL) | | qPCR result (2022) | | Manual survey result (2021/2022) |
|---|---|---|---|---|---|---|---|
| | 0.22µm | 0.45µm | 0.22µm | 0.45µm | 0.22µm | 0.45µm | |
| **EASTERN SECTION DITCH** | | | | | | | |
| 1–3 pool | 160 | 285 | 30.10 | 40.90 | 0/12 | 0/12 | Not found |
| 4 | 250 | 400 | 8.54 | 7.04 | 0/12 | 0/12 | Not found |
| 5 | 350 | 300 | 14.80 | 14.50 | 0/12 | 0/12 | Not found |
| 6 | 350 | 300 | 8.49 | 16.30 | 0/12 | 4/12 | Present (juvenile) |
| 8 | 300 | 250 | 2.40 | 3.08 | 5/12 | 9/12 | Present |
| 8 to 18 lower side | 400 | 300 | 8.91 | 2.67 | 0/12 | 1/12, 2/12, 0/12[#] | Present |
| 18 lower side | 300 | 300 | 9.64 | 12.40 | 0/12 | 11/12 | Present |
| 18 upper side | 400 | 400 | 17.80 | 15.70 | 0/12 | 0/12 | Present |
| **WESTERN SECTION DITCH** | | | | | | | |
| 9 | If 150 | 200 | 11.60 | 5.19 | 0/12 | 0/12 | Not found |
| Coffer to 9 | 300 | 400 | 32.10 | 55.00 | 0/12 | 1/12, 0/12, 0/12[#*] | Not found |
| 11 opposite ditch C | 325 | 300 | 38.50 | 35.00 | 0/12 | 0/12 | Not found |
| Coffer to 11 | 250 | 350 | 14.10 | 32.90 | 0/12 | 0/12 | Not found |
| **SIDE DITCHES** | | | | | | | |
| Coffer to ditch G | 200 | 200 | 27.10 | 28.30 | 0/12 | 0/12 | Not found |
| Coffer to 12 | 200 | 200 | 20.10 | 27.00 | 0/12 | 0/12 | Present |
| Ditch H upper section | 150 | 200 | 12.10 | 0.848 | 4/12 | 1/12, 1/12, 1/12[#] | Not found[¥] |
| Coffer to ditch H | 300 | 250 | 54.00 | 9.40 | 0/12 | 0/12 | Not found (operculum)[≠] |
| Coffer to 7 | 200 | 200 | 32.40 | 8.54 | 1/12, 0/12, 2/12[#] | 12/12 | Present |
| 4-J (between 4 and ditch J) | 200 | 200 | 20.90 | 7.92 | 1/12, 0/12, 0/12[#*] | 0/12 | Not found |

Green highlighted samples were found to be positive for *C. chinensis* by both qPCR and manual survey; orange highlighted surveys were found to be positive for *C. chinensis* by qPCR but not manual survey; blue highlighted samples were found to be positive for *C. chinensis* by manual survey but not by qPCR. 0.22µm refers to the Sterivex filter and 0.45µm refers to the PES-GF filter combination.

[#] qPCR was repeated for these samples to confirm positivity.

[*] Samples found to be negative on both retests and are therefore assumed to be negative.

[≠] in April 2022, the only evidence of *C. chinensis* presence was the finding of a snail operculum which may originate from by before the ditch was dammed.

[¥] this site was heavily over-shaded by brambles and have no obvious drainage towards the Glynleigh sewer. No snails were found in April 2022.

## Occupancy analysis

Using the observed percentage agreement of the two methods of 0.80 (1 = 100%) and the probability of random agreement of 0.52, Cohen's kappa coefficient was calculated as 0.59 (95% CI 0.31 to 0.88) for manual survey-positive ponds versus their qPCR analysis results i.e. moderate agreement (combining all samples collected in 2021 and 2022).

Site modelling was used to calculate the occupancy estimate and the probability of detection of *C. chinensis*. Using the combined manual survey and qPCR assay results for the Pevensey Levels in each of 2021 and 2022 the occupancy estimates were calculated (Table 3). Occupancy estimates were 0.58 or 0.47 with a probability of detection of 0.10 or 0.13 respectively (Table 3).

## Discussion

### Assay design

This work was undertaken to develop a species-specific qPCR assay to monitor the presence of *C. chinensis* in the UK and to compare this to manual survey data. The assay, targeting a 204 bp fragment of the COI region of *C. chinensis* was developed and optimised for determining

**Table 3. Parameter estimates for a combination of manual survey and qPCR methods using a single-season model Ψ(-), p(-), that is, assuming constant occupancy and detection.**

| Model | N | -2 log likelihood | Ψ (95% CRI) | Est. P (95% CRI) | SE (P) |
|---|---|---|---|---|---|
| Manual survey plus qPCR (Pevensey Levels 2021) | 2 | 79.36 | 0.58 (0.38, 0.76) | 0.74 (0.52, 0.88) | 0.10 |
| Manual survey plus qPCR (Pevensey Levels 2022) | 2 | 39.47 | 0.47 (0.25, 0.71) | 0.77 (0.44, 0.93) | 0.13 |

Where N = number of parameters, Ψ = occupancy estimate, P = estimated detection rate. Sample size = 36 sites (Pevensey Levels 2021), 18 sites (Pevensey Levels 2022).

the presence of *C. chinensis* based on the detection of eDNA in ditch water samples at the Pevensey Levels, UK. eDNA-based assays require that closely related and co-occurring species whose DNA may be present in the environmental samples are considered when designing the assay to ensure their specificity [37]. The qPCR designed in this study successfully distinguished between *C. chinensis* and other co-occurring species in that no cross-amplification was detected either *in silico* or *in vitro*. The only exception to this was *C. cathayensis* a very closely related species found to cross-amplify during *in silico* testing. *C. chinensis* and *C. cathayensis* are distinguishable via various anatomical characteristics [18] but have very similar COI sequences which prevented the development of species-specific primers. We were unable to source DNA from this species to test our qPCR assay *in vitro*, however, as the species is not present in the UK this does not affect the application of the qPCR assay. A selection of positive amplifications were sequenced to confirm the species identity as *C. chinensis laeta* at the site. If *C. cathayensis* was introduced into the UK this assay may also be suitable for its detection and any positive amplifications could be confirmed by sequencing.

## Pevensey Levels

Manual survey work carried out in July 2019 found the presence of a recruiting population of *C. chinensis* in an approximately 400 m stretch of the Pevensey Levels ditch system. Further surveys carried out at this ditch system in February 2021 found a number of smaller juveniles which suggested that a breeding population was present (Willing 2021a [unpublished]). Confirmation of the breeding population was made during August 2021 when several *C. chinensis* specimens were collected and returned to the laboratory for use as positive control material. During tissue removal for DNA extraction, one of the specimens was found to contain juvenile snails—this species is known to give live birth during June to October potentially having more than 160 young in their lifetime [38].

In August 2021, additional manual survey work carried out following eDNA sample collection, showed that *C. chinensis* had colonised an additional 138 m of ditch (downstream) since it was surveyed in February 2021 (Willing 2021b [unpublished]). The August 2021 eDNA survey confirmed the presence of *C. chinensis* in these areas, but also identified five other areas where *C. chinensis* eDNA may be present despite not being found during manual survey (Rees et al. 2022) (Fig 2). The opposite was true for four additional sites where *C. chinensis* was found by manual survey but not during eDNA survey.

Site occupancy models can be used to account for imperfect detection and were used by Schmidt et al. [39] to demonstrate their applicability to eDNA surveys. When applied to the Pevensey Levels data within this study, site occupancy estimates were greater than the actual observed proportion though not significantly when combining manual survey with eDNA assay. This matches the observed increase in positive detections from 15/36 to 16/36 sample sites in 2021, and 6/18 to 7/18 sample sites in 2022 when both techniques were combined.

Following these combined findings, three coffer dams were installed to prevent further spread of *C. chinensis* within the ditch system (Fig 3). A concern is if *C. chinensis* spreads from

its original source then there is the risk of disruption to the diverse and extensive freshwater ditch ecosystems on the Pevensey Levels, a habitat for many rare invertebrate species including the rare freshwater Mollusc the little whirlpool ram's-horn Snail (*Anisus vorticulus*).

The three erected coffer dams (1–3) were made of metal sheet pile construction. In April 2022, further manual searches found no live *C. chinensis* in ditch sectors lying beyond (outside) any of the coffer dams or in any of the other side channels surveyed (side ditches B, C, D, G, H and J) (Willing 2022 [unpublished]). Therefore, Natural England commissioned a further eDNA and to confirm these results during August 2022.

Water samples taken by Natural England and RSK ADAS staff in August 2022 confirmed the continued presence of *C. chinesis* within the ditch system albeit in fewer sites than in 2021. It was also found that the 0.45 μm PES filter in combination with a 3.1 μm glass fibre pre-filter returned a higher number of positive qPCR results (six samples were positive for *C. chinensis* DNA) than for those samples collected using a 0.22 μm sterivex filter (two samples were positive for *C. chinensis* DNA). Small pore size filters (0.20 to 1.5 μm) have been shown to yield the most eDNA but are prone to clogging when sampling turbid water (water quality was noted to be poor during sample collection) or where there are algal blooms [40–42]. In such conditions, either larger pore-size filter or pre-filtration can be used as was the case with the PES/GF filter combination. Pre-filtration is known to increase single species detection probability [43] and give more consistent results for community compositions. However, the use of pre-filters can lower the DNA yield and the number of detected taxa [44]. In the current study eDNA was extracted from both the pre-filter and the main filter to minimise any DNA loses [45]. A larger pore size filter allows an increased volume of water to be filtered but there is a trade-off in that the smallest particles containing eDNA may pass through the filter. In this case similar volumes of water were filterable through the Sterivex and PES/GF filters.

*C. chinensis* DNA was not found in or around the coffer dam erected between sites 9 and 11 but was found within side ditch H–upper section. Although the addition of the three coffer dams could partly explain this for the side ditches (except H which was still positive for *C. chinensis* DNA), the water quality was poor and there had been a period of very hot weather prior to the 2022 sampling. This caused water levels to drop considerably compared with 2021 levels leaving significant areas of exposed mud at all sampled sites where lots of *C. chinensis* shells were found. It is known that *C. chinensis* favour marginal areas during the summer months, after which the snails migrate to deeper waters for the winter months (favouring the deeper more central areas) [14, 15]. This suggests that the *C. chinensis* population could have been reduced compared with previous years thus reducing the amount of *C. chinensis* DNA present. The concentration of eDNA in a water body will depend on the rate of production versus how long it persists in the environment [46]. The rate of eDNA production for a species will depend on many factors including: the number of individuals present, their physiology and metabolism; and temperature [47]. eDNA is likely to have degraded at a faster rate due to the hot weather and the high UV levels associated with this hot weather [46–48] Pilliod et al. 2014 [7] which could help to explain why sites which were positive for *C. chinensis* in 2021 were negative for *C. chinensis* in 2022.

During the 2021 surveys it was discovered that a new shallow water scrape had recently been created in a field lying on the far western side at the outflow end of the infested ditch (Fig 2). The scrape was not surveyed due to access issues and with no apparent ditch connection with the infested ditch colonisation by *C. chinensis* seems unlikely, however if the snail was accidentally (or intentionally) introduced, the shallow water conditions are likely to suit it.

## Conclusions

This study has shown that eDNA sampling combined with manual surveys has allowed Natural England to determine the spread of *C. chinenis* in the ditch network at the Pevensey Levels. The results gave confidence that the placement of the three coffer dams were in the right location and a fourth coffer dam, planned for 2023, is to be placed in side ditch J. The implications of the long-term use of coffer dams will need to be considered as there is a concern over the possibility of the site flooding in the winter and whether water levels can be managed long-term. Natural England are planning to dredge the ditch over the next few years and eDNA sampling will be invaluable in determining the success of this operation especially if the snails are in low numbers. The new shallow scrape that was found to be suitable for colonisation by *C. chinensis* (Willing 2022 [unpublished]) should also be surveyed by both manual and eDNA survey.

## Supporting information

**S1 File. Ditch water and snail sample information.**
(DOCX)

**S2 File. qPCR information.**
(DOCX)

## Acknowledgments

We would like to thank the Natural England Area Team staff for assisting in collection of the water samples and Dr Martin Willing (conchologist) for *C. chinensis* survey confirming the spread of the species further downstream in August 2021.

## Author Contributions

**Conceptualization:** Gavin H. Measures.

**Funding acquisition:** Helen C. Rees.

**Investigation:** Helen C. Rees.

**Methodology:** Helen C. Rees, Steven D. Kane, Ben C. Maddison.

**Project administration:** Helen C. Rees, Gavin H. Measures.

**Supervision:** Helen C. Rees.

**Writing – original draft:** Helen C. Rees.

**Writing – review & editing:** Helen C. Rees, Gavin H. Measures, Ben C. Maddison.

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
