## [Decision Letter · Decision Letter 0]

26 Jul 2023

PONE-D-23-17026qPCR assay for the specific detection of the Chinese mystery snail (Cipangopaludina chinensis) in the UK.PLOS ONE

Dear Dr. Rees,

Thank you for submitting your manuscript to PLOS ONE. After careful consideration, we feel that it has merit but does not fully meet PLOS ONE’s publication criteria as it currently stands. Therefore, we invite you to submit a revised version of the manuscript that addresses the points raised during the review process.

We look forward to receiving your revised manuscript.

Kind regards,

Hudson Alves Pinto, Ph.D

Academic Editor

PLOS ONE

https://besjournals.onlinelibrary.wiley.com/doi/full/10.1111/1365-2664.12306

In your revision ensure you cite all your sources (including your own works), and quote or rephrase any duplicated text outside the methods section. Further consideration is dependent on these concerns being addressed.

3. We note that you have referenced

(

Measures, G.H., (2022). A survey to assess the presence of the Chinese Mystery Snail (Cipangopaludina chinensis) at Southampton Common SSSI - 24th – 25th August 2022. Natural England (unpublished report).

Willing, M., (2021a). A survey to assess the status of the Chinese Mystery Snail Cipangopaludina chinensis & the Little Whirlpool Ram’s-horn snail Anisus vorticulus in a ditch and it’s connecting channels on Glynleigh Level, Pevensey Levels (February – March 2021). Report to Natural England (unpublished).

Willing, M. (2021b). Appendix Notes to accompany: A survey to assess the status of the Chinese Mystery Snail Cipangopaludina chinensis & the Little Whirlpool Ram’s-horn snail Anisus vorticulus in a ditch & it’s connecting channels on Glynleigh Level, Pevensey Levels (February – March 2021). August 2021. Report to Natural England (unpublished).

Willing, M. (2022). A survey to assess the presence of the Chinese Mystery Snail Cipangopaludina chinensis in relation to the recent placement of three coffer dams and resurvey of a site supporting the Little Whirlpool Ram’s-horn snail Anisus vorticulus on Glynleigh Level, Pevensey Levels, April 2022. Report to Natural England (unpublished). 

)

which has currently not yet been accepted for publication. Please remove this from your References and amend this to state in the body of your manuscript: (ie “Bewick et al. [Unpublished]”) as detailed online in our guide for authors

5. We note that Figures 2 and 3 in your submission contain [map/satellite] images which may be copyrighted. All PLOS content is published under the Creative Commons Attribution License (CC BY 4.0), which means that the manuscript, images, and Supporting Information files will be freely available online, and any third party is permitted to access, download, copy, distribute, and use these materials in any way, even commercially, with proper attribution. For these reasons, we cannot publish previously copyrighted maps or satellite images created using proprietary data, such as Google software (Google Maps, Street View, and Earth). For more information, see our copyright guidelines: http://journals.plos.org/plosone/s/licenses-and-copyright.

a. You may seek permission from the original copyright holder of Figures 2 and 3 to publish the content specifically under the CC BY 4.0 license. 

Additional Editor Comments:

The Reviewers presented very detailed reports, which I fell that will contribute a lot to improviment of the final version of the MS. Please, take special attention of Reviewer 2 on the sequencing the samples of eDNA. This is important to support the specificity of the proposed assay.

I look forward to receive the corrected version of this interesting MS.

Reviewers' comments:

Reviewer's Responses to Questions

**Comments to the Author**

1. Is the manuscript technically sound, and do the data support the conclusions?

Reviewer #1: Yes

Reviewer #2: Yes

2. Has the statistical analysis been performed appropriately and rigorously? 

Reviewer #1: Yes

Reviewer #2: Yes

3. Have the authors made all data underlying the findings in their manuscript fully available?

Reviewer #1: Yes

Reviewer #2: Yes

4. Is the manuscript presented in an intelligible fashion and written in standard English?

Reviewer #1: Yes

Reviewer #2: Yes

5. Review Comments to the Author

Reviewer #1: Dear Authors,

I have carefully reviewed the paper titled 'qPCR assay for the specific detection of the Chinese mystery snail (Cipangopaludina chinensis) in the UK', and I would like to provide feedback on behalf of the reviewing committee.

General Comments:

The paper is well-written and the experiments have been conducted in a rigorous manner. The authors should be commended for their efforts in addressing the research problem.

Specific Feedback:

I have attached a document containing minor observations, suggestions, and corrections to enhance the clarity and precision of the paper. Please review the document and consider the proposed changes.

Please note that the comments provided are intended to improve the overall quality and impact of your research.

Thank you for considering my feedback. I look forward to seeing the revised version of your paper.

Reviewer #2: I thank the authors for their time spent in the field and lab to develop this assay. Such assays are necessary for the future monitoring of invasive species and their locations and helps add to the body of work and field of eDNA assays. I recommend publication with minor revisions assuming that comments about line 304 are clarified.

Line 13: I think the year is off by 1 year. Gray 1834 instead of 1833 – at least that is what is in the citation.

Line 94-96: Check for typos and grammar. Also, common names are not capitalized (with location exceptions). I have noticed that there are typos in the manuscript with a few instances of the species not being italicized (be sure to check your figure legends).

Line 121: For every site visited?

What is ADAS and how many staff were there doing this work?

Typically, the catalog number of associated filters is stated so someone else could buy the exact filters to repeat the assay.

Line 140: What is ATL buffer? What concentration of proteinase K?

Line 196: Including a supplementary table of the species chosen would be helpful as other researchers may want to see what species those primers would likely not amplify. Were any of these the sister species? How did you determine what was a closely related species? Please include a citation?

Line 263: Species needs to be in italics

Line 304: Does this mean that you all sequenced the positive samples that came from the qPCR assay and the results were the correct species? If so, I think that shows that is assay is working well and it needs to go in your methods and results section. If not, then I would recommend going back to those positive eDNA samples and sequencing them to be sure you are not amplifying another organism. Quite often in real-world sampling (field samples) I have noticed eDNA assays go out the “window” because they were amplifying the wrong target. Sometimes it is not even closely related to the target organism.

Additional Notes:

For the map figures I suggest making a map that shows 1) where eDNA samples were taken and if they were positive 2) where snails were collected and if they were positive for C. chinensis. Maybe a pie chart graphic or something to make it clear where the snails were found and if the eDNA assay was positive from that site. From my point of view it is less concerning to find positive eDNA, but no snails compared to negative eDNA and finding snails. It would suggest that the assay is not sensitive enough. There are times where you might not be able to find the organism, but rather “traces” (eDNA) of the organism. The authors made the point that there is room for improvement, but I think what you have shown suggests there is room for improvement when it comes to sensitivity assuming the authors sequenced the positive qPCR samples.

When writing a paper about the development of an assay every detail (even if it seems extra) matters to someone who would want to repeat the assay in future. I also suggest adding more details in the materials and methods to make it clearer people who may use this assay in the future.

The locations/names of the sites the authors visited might be clear to the authors, but not clear to someone else who may also want to visit these sites. Please include GPS coordinates in your tables.

6. PLOS authors have the option to publish the peer review history of their article (what does this mean?). If published, this will include your full peer review and any attached files.

Reviewer #1: No

Reviewer #2: No

---

## [Editor Report · Decision Letter 1]

15 Sep 2023

Quantitative PCR (qPCR) assay for the specific detection of the Chinese mystery snail (Cipangopaludina chinensis) in the UK.

PONE-D-23-17026R1

Dear Dr. Rees,

We’re pleased to inform you that your manuscript has been judged scientifically suitable for publication and will be formally accepted for publication once it meets all outstanding technical requirements.

Kind regards,

Hudson Alves Pinto, Ph.D

Academic Editor

PLOS ONE

Additional Editor Comments (optional):

The MS improved a lot and it is now suitable for publication. I congrats the authors by the so detailed correction.
---

## [Editor Report · Acceptance letter]

26 Sep 2023

PONE-D-23-17026R1 

Quantitative PCR (qPCR) assay for the specific detection of the Chinese mystery snail (*Cipangopaludina chinensis*) in the UK. 

Dear Dr. Rees:

I'm pleased to inform you that your manuscript has been deemed suitable for publication in PLOS ONE. Congratulations! Your manuscript is now with our production department. 

Kind regards, 

on behalf of

Dr. Hudson Alves Pinto 

Academic Editor

PLOS ONE